# Imaging fascicular organization of rat sciatic nerves with fast neural electrical impedance tomography

Enrico Ravagli [1,5], Svetlana Mastitskaya [1,5✉], Nicole Thompson [1,5], Francesco Iacoviello [2], Paul R. Shearing[2], Justin Perkins[3], Alexander V. Gourine [4], Kirill Aristovich[1] & David Holder[1]

Imaging compound action potentials (CAPs) in peripheral nerves could help avoid side effects in neuromodulation by selective stimulation of identified fascicles. Existing methods have low resolution, limited imaging depth, or are invasive. Fast neural electrical impedance tomography (EIT) allows fascicular CAP imaging with a resolution of <200 µm, <1 ms using a non-penetrating flexible nerve cuff electrode array. Here, we validate EIT imaging in rat sciatic nerve by comparison to micro-computed tomography (microCT) and histology with fluorescent dextran tracers. With EIT, there are reproducible localized changes in tissue impedance in response to stimulation of individual fascicles (tibial, peroneal and sural). The reconstructed EIT images correspond to microCT scans and histology, with significant separation between the fascicles (p < 0.01). The mean fascicle position is identified with an accuracy of 6% of nerve diameter. This suggests fast neural EIT can reliably image the functional fascicular anatomy of the nerves and so aid selective neuromodulation.

[1] Department of Medical Physics and Biomedical Engineering, University College London, London, UK. [2] Electrochemical Innovation Laboratory, Department of Chemical Engineering, University College London, London, UK. [3] Clinical Science and Services, Royal Veterinary College, Hawkshead Lane, Hatfield, UK. [4] Centre for Cardiovascular and Metabolic Neuroscience, Department of Neuroscience, Physiology and Pharmacology, University College London, London, UK. [5]These authors contributed equally: Enrico Ravagli, Svetlana Mastitskaya, Nicole Thompson. ✉email: s.mastitskaya@ucl.ac.uk

Bioelectronic medicine, or Electroceuticals, is a rapidly developing therapeutic field in which biomedical devices are used electrically to block, record, or stimulate neural activity as an alternative to drugs. A prime target for such interventions is the cervical vagus nerve, which innervates various visceral organs and muscles, including the pharynx, larynx, heart, lungs, and gastrointestinal tract. The human cervical vagus nerve comprises about 5–8 fascicles[1], but their anatomical relation to supplied organs is largely unknown. Until now, therapeutic electrical stimulation of the vagus nerve (VNS, vagus nerve stimulation) has been of the entire nerve. This may result in the activation of unwanted organs and cause side effects, which limits therapeutic opportunities. In principle, this could be avoided by selective stimulation of specific fascicles, but this is currently not possible due to the lack of understanding of fascicular function.

Currently, there is no technique that allows for non-invasive imaging and tracing of organ-specific projections of the vagus nerve in large mammals. Neural tracing allows labeling of the cell bodies of neurons projecting to an innervated organ, but this may not label the axons[2] and requires postmortem examination with histology and serial computerized stitching and tracking. Optical coherence tomography, photoacoustic tomography, and magnetic resonance imaging have low resolution and penetration depth[3], and multielectrode arrays are damaging to the nerve[4]. High-resolution ultrasound provides a good axial resolution of up to 400 μm but has a low soft tissue contrast, which does not enable differentiation between fine nerve structures such as the fascicles and intrafascicular perineurium[5]. Magnetic resonance neurography provides good soft tissue contrast but has a spatial resolution of 0.4–0.8 mm, which is insufficient for imaging of fine fascicular structures. This may be increased by imaging small fields of view and applying high magnetic fields but then there is a restricted field of view[6]. Phase contrast radiography demonstrates the fascicular nerve architecture[7] but does not enable tracking of fascicular functional connections over tens of cm.

Potential ex vivo methods for imaging the fascicular anatomy of nerves include histology and micro-computed tomography (microCT). Histological examination after appropriate staining may be used to study the microanatomy of tissues and trace functional connections of nerve fascicles to their end organs[8]. It requires fixation of the tissue, staining and microscopy with computerized tracking of numerous serial sections. It is accepted as a gold standard technique but is time-consuming and may be prone to artifacts. MicroCT utilizes X-rays and provides tomographic imaging with a spatial resolution up to 4 μm in three dimensions with little distortion of the sample and minimal artifacts[9]. For application to imaging of soft tissues, including nerves, it requires the use of X-ray contrast agents such as Lugol's iodine solution, which cannot be used in vivo. The method has previously been applied to nerves[10,11]. However, these required phase contrast scanners and long preprocessing, which was not practical for tracking of long segments of nerve needed for delineation of functional anatomy of fascicles and their projections to the end organs. The method has been optimized in our group[12]. Using a conventional microCT scanner, it was possible to image fascicles in rat sciatic and pig vagus nerve to a length of 4 cm with a resolution of 4 μm.

Neural tracing followed by histological examination has been successfully used to study neural connections within the central and peripheral nervous system (PNS). Usually, in the PNS, the tracer is applied to the peripheral organ or tissue. The majority of tracers are rapidly transported to the cell bodies of projecting neurons and do not easily allow labeling of the axon shafts (e.g., horse radish peroxidase, FluoroGold (FG), FastBlue, Diamidino Yellow)[2]. To our knowledge, there is no systematic study assessing the efficiency of neural tracers for axonal labeling at the mid-shaft of the nerve. A suitable tracer for fascicular labeling and end organ tracing should not be neurotoxic and allow practicable labeling at a defined time. Tracer toxicity appears to be poorly understood. In rat sciatic nerve, toxicity and effect on motor function was studied for FG, TrueBlue (TB), and Fluoro-Rubi (FR)[13]. Both FG and TB caused functional impairment and axonal degeneration, which were only reversible by week 4 (TB) or week 24 (FG) post-injection, whereas FR (which is a dextran conjugate) did not cause functional deficits and was comparable to vehicle injection[13]. Dextran conjugates are hydrophilic polysaccharides, non-toxic, and relatively inert; they are transported by passive diffusion[14] and rapidly fill up the entire length of the axon when injected into the distal end of the nerve. Other tracers able to reliably label the axons without affecting the compound action potential (CAP) are adeno-associated viral vectors (AAVs). However, AAVs require a long incubation time and are mostly efficient for anterograde tracing only (from cell body to the nerve terminal)[15]. Towne et al.[16] successfully traced the motor neurons in rat sciatic nerve using AAV6-mediated delivery of fluorescent reporter gene into motor neurons—the efficiency of retrograde transport reached 80–90% after intra-muscular injections of the viral vector into gastrocnemius or tibialis anterior muscle. However, to achieve successful transfection, the injections had to be made in neonates, and incubation time took four weeks.

Electrical impedance tomography (EIT) is a non-invasive technique that allows imaging of variations in electrical impedance inside a volume of interest. It is achieved by mathematical reconstruction of electrical impedance data collected from an array of external electrodes[17]. Its clinical utility has been demonstrated for monitoring lung function[18], and research is in progress into its potential use in breast cancer[19], stroke[20,21], and detection of epileptic seizure onset zones[22]. It can also serve as a means to produce high-resolution tomographic images of activity in excitable neural tissue in brain and nerve with a millisecond and submillimeter resolution, "fast neural EIT." Our group has developed this for imaging neuronal depolarization in the brain during normal and epileptic activity with a resolution of 1 ms and <200 μm in the rat cerebral cortex[23,24]. The principle is that impedance in neuronal membranes falls as ion channels open during evoked activity[17]. This produces a decrease in the bulk electrical impedance of ~0.1% during neuronal depolarization, which allows the applied EIT current to pass into the intracellular space, whereas at rest the EIT current predominantly travels in the extracellular space[17].

In fast neural EIT of peripheral nerve, the changes of electrical impedance are imaged with a cuff-like circumferential array of electrodes placed around the nerve[25]. Images are reconstructed from a set of transfer impedances. These are collected by applying a sub-threshold alternating current of ~50 μA at ~6 kHz constant current to two out of 14 electrodes placed near-diametrically, generally ~100° apart; 12 voltages from all other available electrodes in the ring are recorded. Typically, impedance epochs are produced over 200 ms by averaging during 300 CAPs at 5 Hz elicited by supramaximal stimulation of a peripheral branch of the nerve over 60 s. A full dataset for image reconstruction is then collected by sequential switching between all possible 14 electrode pairs with the same spacing, using electronic multiplexers (total time 14 min per branch/fascicle). The applied frequency of 6 kHz provides the optimal balance between contamination with endogenous potentials at lower frequencies and a decrease in the impedance response to depolarization at higher frequencies, both empirically and by biophysical modeling[26,27]. The resulting dataset allows imaging of neuronal depolarization with a high spatio-temporal resolution of <1 ms and 200 μm in rat sciatic nerve[25,28]. Optimization of related technical aspects has included electrode design and fabrication[29], current injection protocols[28],

and EIT frequency range[30]. The data in this study were collected with the "UCL ScouseTom" EIT system[31]. This comprises a commercial high-specification electroencephalogram (EEG) recorder able to record potentials in parallel over up to 128 channels, a commercial low-noise constant current generator and a custom multiplexer and controlling software.

Fast neural EIT has unique potential for identification and localization of activity within peripheral nerves. This could help to delineate functional fascicular anatomy, by indicating specific fascicles active in synchrony with individual organs or their supplying peripheral branches. The vision is that it could be left chronically in situ and allow imaging of fascicles of interest by gating to physiological organ activity (e.g., the ECG for heart, respiration for lung, or electrogastrogram for stomach). A pair of second nerve cuffs with identical geometry could then be used for selective stimulation and avoidance of off-target effects. It could also have practical applications in reconstructive nerve surgery as a method for functional tractography for correct fascicular repair[32] and in human robotics[33,34].

The method of fast neural EIT applied to peripheral nerve has been published but was a technical illustration of the method in the sciatic nerve without independent validation and only of two fascicles—the posterior tibial and peroneal[25,28]. In this work, we present the first study in which the final optimized method was independently validated against histology with fluorescent dextran neural tracers and microCT. This was achieved in all three main fascicles in the rat sciatic nerve. We chose the model of rat sciatic nerve for our study because of its clear somatotopic organization. Compared to the human sciatic nerve, where tibial, peroneal, and sural branches consist of fascicular groups, not individual fascicles[35], the rat sciatic nerve is a simpler model of peripheral nerve. In rats, both afferent and efferent nerve fibers at the level of common sciatic nerve are organized into three main fascicles. These fascicles give rise to sciatic, peroneal, and sural branches[36]. A secondary purpose was to refine the gold standard methods of neural tracer histology and microCT to be suitable for this comparison. These are mature techniques, but some development was needed to apply them successfully to fascicle tracking over several cm.

The CAPs in three main fascicles of the sciatic nerve in the thigh of anesthetized Sprague-Dawley rats were imaged with fast neural EIT, using a cylindrical cuff with 14 electrodes arranged in a ring and supramaximal electrical stimulation of the tibial, peroneal, and sural peripheral branches (Fig. 1). To select a robust method for neural tracing, in preliminary studies we evaluated eight different tracers, three conventional fluorescent tracers, and five viral vectors. FG and 1,1′-dioctadecyl-3,3,3′3′-tetramethyl-indocarbocyanine perchlorate allowed reproducible labeling of the fascicles but impaired evoked CAPs, which obviated reproducible EIT data. Five AAVs tested did not produce reliable results at 5–6 weeks after injection, whereas dextran tracers were reliable from the outset. We therefore opted for fluorescent dextran conjugates. These are non-toxic sugar-based tracers that travel reliably proximally in the nerve by passive diffusion, and application of these tracers did not reduce the CAPs. As only two colors were available, this was applied to the posterior tibial and peroneal fascicles. On histological examination, the third (sural) fascicle was identified as the main unstained fascicle. The experimental timeline was as follows:

- Forty-eight hours before the EIT experiment, the animal received fluorescent dextran tracer injections into the tibial and peroneal branches (Fig. 1a).
- EIT experiment: tibial, peroneal, and sural branches were electrically stimulated supramaximally at 5 Hz. EIT was recorded using a cuff electrode array placed around the sciatic nerve in the thigh (Fig. 1b).

- Postmortem, the nerve was removed and the entire length of several cm was imaged with microCT in order to identify the origin of each fascicle in the thigh portion of the nerve (Fig. 1c) and processed for histology (Fig. 1d).
- Resulting images from EIT, microCT, and histological slices were co-registered for statistical analysis (Fig. 1e).

Here, we show the reliability of fast neural EIT as a non-invasive method for imaging fascicular anatomy of peripheral nerves. This novel technique is fully validated against the existing gold standard methods, namely, neural tracer histology and microCT. EIT has a significant advantage over other in vivo neural imaging techniques as it allows for imaging of functional activity and thus can aid selective neuromodulation.

## Results

**CAPs and impedance changes in individual fascicles**. The recorded CAPs were $211 \pm 142$, $168 \pm 90$, and $114 \pm 39$ μV (peak impedance changes of $9.7 \pm 5.5$ μV ($0.04 \pm 0.02\%$), $13.7 \pm 15.8$ μV ($0.05 \pm 0.04\%$), and $13.4 \pm 11.3$ μV ($0.05 \pm 0.03\%$)) with onset times of $0.7 \pm 0.2$, $0.8 \pm 0.1$, and $0.8 \pm 0.1$ ms for tibial, peroneal, and sural fascicles, respectively. All were significant with respect to baseline noise (paired $t$ test with respect to inter-stimulus noise, $p < 0.05$) (Fig. 2). The duration above 30% of peak impedance change was $0.7 \pm 0.3$, $0.8 \pm 0.3$, and $0.8 \pm 0.3$ ms for each fascicle, respectively, as above, and each corresponded to the peak-evoked CAP measured on the same electrode. Conduction velocities were 20–60 m/s, which correspond mainly to A and B fibers. The signal-to-noise ratio (SNR) of evoked averaged peak impedance changes was $16.7 \pm 12.4$ or $24.5$ dB at 6 kHz ($n = 15$ fascicles in $N = 5$ nerves).

**Validation techniques and co-registration**. MicroCT imaging allowed for the visualization and clear distinguishability ($d = 0.998$, as in ref. [12]) of the three main fascicles of the rat sciatic nerves and their subsequent segmentation (Fig. 3). The average diameter difference of all three fascicles and the full nerves between microCT and histology was <2.5%. To image the full rat sciatic nerve, 3–4 overlapping scans of ~8 mm length taking 3.5 h each were needed. Images for the three techniques were co-registered by fitting the microCT and neural tracer images to the circular shape of EIT images. This was achieved by rigid deformation and rotational alignment to the fiducial marker of a surgical suture on the nerve cuff (Fig. 4).

The speed of fluorescent dextran diffusion within the nerve was evaluated and incubation time of 48 h was chosen as optimal (Supplementary Note 1). Resulting fluorescent images were of high contrast and specificity, with clearly distinguishable boundaries of the fascicles (peroneal and tibial) at the level of EIT cuff placement (Fig. 4b).

**Comparison of EIT images with microCT and neural tracing**. Three clearly distinct zones of activation due to stimulation of tibial, peroneal, and sural fascicles were apparent in reconstructed EIT images for each nerve (Fig. 4a and Supplementary Fig. 2). All recordings had significant peak δV variations related to evoked activity ($p < 0.05$). Of the 196 available transfer impedances per dataset, some were excluded because of low SNR. Then $146 \pm 10$ δV traces were used for EIT image reconstruction. Visual inspection indicates a close correlation of the three fascicles across different techniques (Figs. 4 and 5 and Supplementary Note 2).

Objective assessment was made using analysis of the radial and angular positions of the center of mass (CoM) of each fascicle. There was significant separation between the fascicles ($p < 0.01$

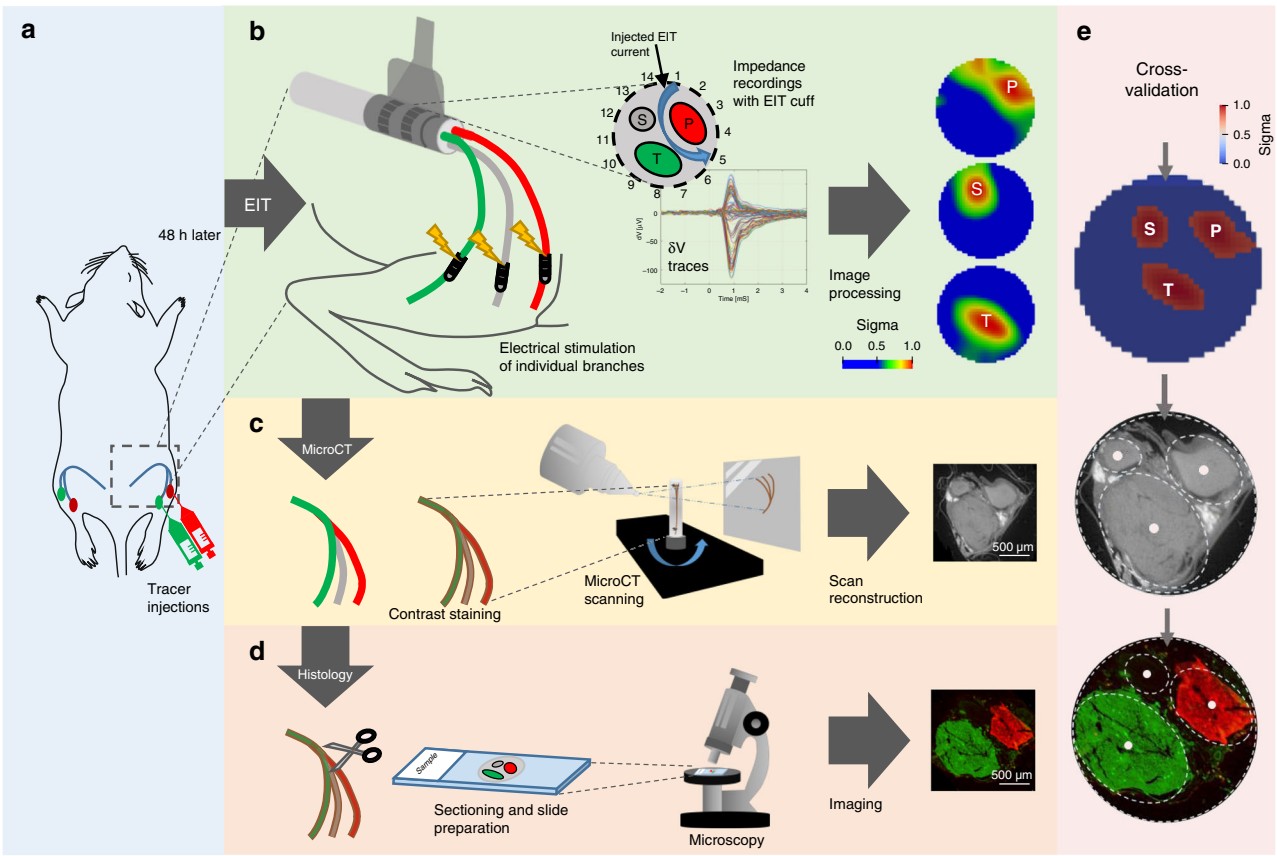

**Fig. 1 Experimental design. a** Neural tracers were injected into the peroneal and tibial fascicles of the rat sciatic nerve 48 h before the EIT experiment. **b** EIT recordings and image reconstruction. Left: peroneal (red), tibial (green), and sural (gray) fascicles were electrically stimulated with bipolar electrodes placed 1–1.5 cm distally from the EIT cuff. Middle: diagram showing the 4-off electrode spacing configuration in the 14-electrode cuff forming a full ring around the rat common sciatic nerve; example of multiple δV traces from an experimental EIT recording. Right: Exemplar cross-sectional localization of the recorded fascicular activity from tibial (T), peroneal (P), and sural (S) fascicles at the level of the common sciatic nerve. The range of values for every image was normalized between 0 and 1 and the rainbow color scale indicates the top 50% of color for each image, i.e., full width at half maximum (FWHM) intensity scaling. **c** After the EIT experiment, the nerve was scanned with microCT. **d** Fluorescent microscopy of histological sections. In this exemplar section, tibial fascicle is labeled with fluorescent Dextran-Alexa 488 (green) and peroneal fascicle with Dextran-Alexa 555 (red); sural fascicle is not labeled. **e** Cross-validation of EIT images of impedance changes in individual fascicles against microCT and histological sections.

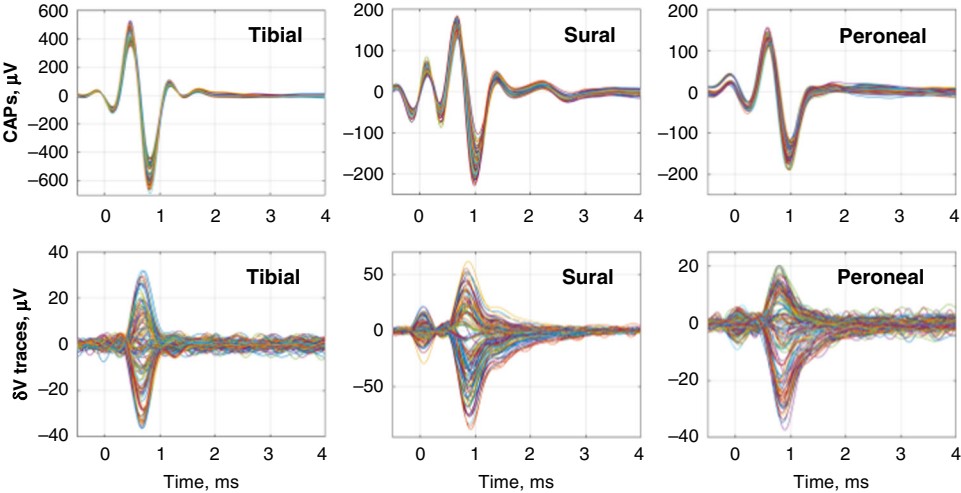

**Fig. 2 Representative compound action potentials (CAPs) and δV traces.** CAPs were recorded on all 14 electrodes of the EIT cuff; δV traces were recorded on 14 electrodes × 14 current injections = 196 for tibial, sural, and peroneal fascicles. Source data are available.

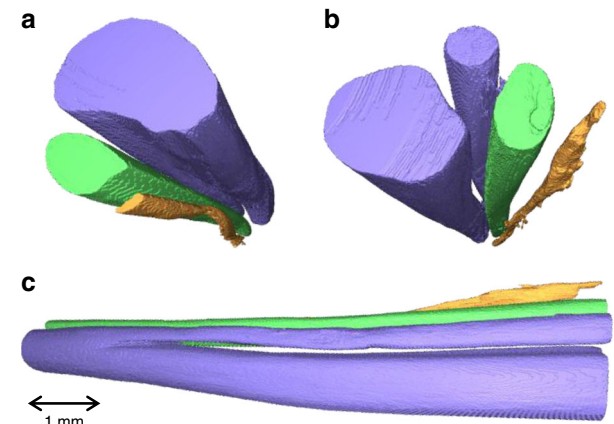

**Fig. 3 Example of microCT segmentation of rat sciatic nerve.** The segmented tibial (purple large), peroneal (green), sural (purple small, branching off tibial), and post-cutaneous (orange) fascicles can be seen from the **a** proximal, **b** distal, and **c** from proximal to distal (left to right) orientations.

for all three fascicles and three techniques, two-way analysis of variance (ANOVA)), and no significant difference between the techniques ($p > 0.05$, two-way ANOVA). Scatter around the mean fascicle position was 86, 66, and 84 µm (6.2, 4.7, and 6.0% of the nerve diameter), for EIT, microCT, and neural tracer histology, respectively (Fig. 5).

## Discussion

Fast neural EIT successfully produced images of localized CAP in peripheral nerve, which corresponded to fascicular activity. Evoked impedance changes had a SNR of 17 [−] for the imaged A and B fast myelinated fibers, applying 6 kHz current. The mean error in fascicle location was 5–6% of nerve diameter and similar for each technique. We also developed microCT and neural tracing with fluorescent dextran conjugates to be suitable for independent validation. These allowed successful identification of individual fascicles at the distal sites of their anatomical projections and reliable tracing proximally to the level of common sciatic nerve where the EIT cuff electrode was positioned. Upon histological examination of the cross-sections at the level of EIT cuff, the fluorescent dextran conjugates were shown to label the

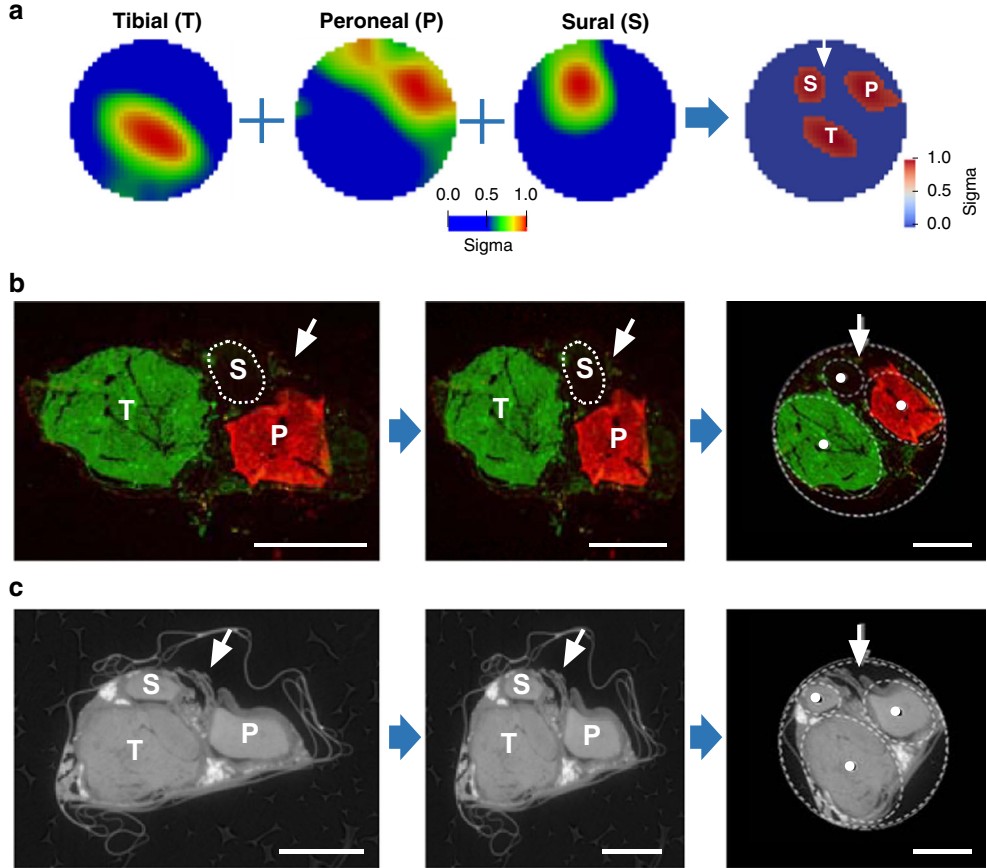

**Fig. 4 Examples of co-registration of fascicle angular location with three techniques. a** Individual images of evoked fascicular activity reconstructed from EIT recordings and the merged image of reconstructed top 10% activations for all three fascicles. All EIT recordings are taken in two repeats for each fascicle to ensure the reproducibility, but just one recording was used for reconstruction. **b** Left: fluorescent image of the cross-section of the common sciatic nerve at the area corresponding to EIT recordings with tibial fascicle labeled with fluorescent Dextran-Alexa 488 (green) and peroneal fascicle with Dextran-Alexa 555 (red). The margins of the sural fascicle that did not receive tracer injection are demarcated with a dashed line. Middle: single-axis rescaling. Right: rotation of histology image for purposes of cross-validation of centers of mass (CoMs) of fascicles between three techniques. The external boundaries of the nerve were fitted to a circular profile after rigid deformation. Each fascicle was fitted into an ellipsoid profile (dashed white line) with a CoM (white dot in the middle), which was further used as a ground truth coordinate in the cross-validation study. Scale bar is 500 µm. **c** Left: representative *XY* plane slice of the microCT scan of the nerve in the area corresponding to EIT recordings with fascicles clearly visible and snippets of the silk thread marking the EIT cuff opening area (pointed at with white arrow). Middle and right: rigid deformation and rotation of microCT image for co-registration. Scale bar is 500 µm.

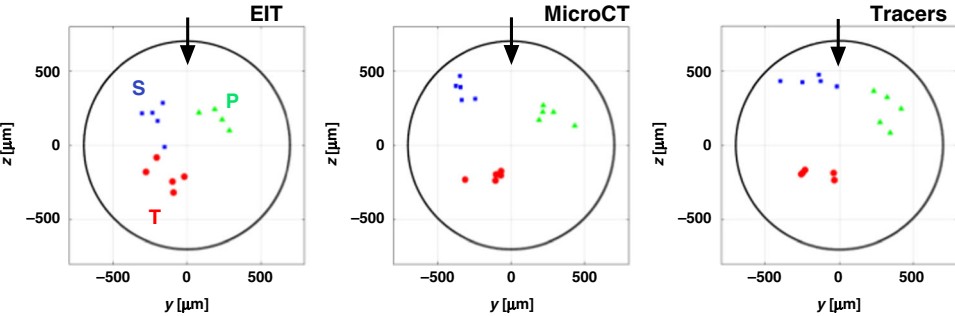

**Fig. 5 Centers of mass (CoMs) of fascicles.** CoMs derived for all three techniques (a total of $N = 5$ nerves in 4 animals, including the right sciatic nerve in 3 animals and both left and right nerves in the fourth animal) oriented to position the fiducial marker (cuff opening) at 12 O'clock (black arrow); red/green/ blue dots for tibial, peroneal, and sural fascicles, respectively. The images from the single left-sided nerve were flipped in the horizontal plane to allow for combination of all nerves in the same dataset. The nerve boundary is shown as a black circle. There was significant separation between the fascicles ($p < 0.01$, two-way ANOVA). Source data are available.

entire target fascicle and clearly demarcate the boundaries of the nerve (no fluorescence observed in epineurium).

Fast neural EIT in the cerebral cortex has been validated by comparison to the independent techniques of local field potentials, current source density, and intrinsic optical activity[37]. Here, we present the first validation of fast neural EIT in peripheral nerve. EIT is a soft-field imaging modality, with the resulting images having no sharp borders due to the type of image reconstruction process. In this specific application of fast neural EIT, the sharp edges from the real fascicles are blurred into the smooth transitions with the kernel size ~100 μm and so typically fast neural EIT has a spatial resolution of ~10% of the image diameter. It is likely to provide a unique localized image of neuronal depolarization, but it is not necessarily restricted to individual fascicles. The other two methods were chosen in order to demonstrate a correspondence with the localized fascicular activity, which might be expected on physiological grounds. The chosen independent validation methods also have intrinsic geometric errors, which necessarily result from the tissue fixation procedure, consequent deformation of the tissue, and the need to deform the resulting images for co-registration. The overall mean percentage shrinkage of formalin-fixed tissue compared to the fresh specimen, for example, is ~10% but can vary depending on tissue type and fixation duration. For example, shrinkage of solid tumor tissue specimens varies between 4.6 and 11.2%[38], whereas the longitudinal shrinkage of an optic nerve after formalin fixation was reported to reach 30%, and this value varied with the age of patients (higher with younger age)[39]. Nevertheless, there appeared to be a good correlation between fascicles and techniques within ~10% of nerve diameter, which is reasonable for typical experimental error[37].

EIT and inverse source analysis (ISA) are mathematically similar but EIT has several advantages: unlike in ISA, the electric source in EIT is known—thus there is a unique inverse solution; EIT allows collection of more independent data: for $n$ electrodes, there are $O(n^2)$ independent measurements, compared to $O(n)$ measurements for ISA. Another significant advantage of EIT over ISA is that it has no field cancellation problem: impedance changes due to ion channels opening are highly unimodal and so any cancellation of signals with EIT is negligible, whereas, in ISA, positive and negative fields may cancel without reaching surface electrodes[22]. Early attempts at fascicle localization with ISA achieved only partial success. Activity in multiple pathways could only be successfully localized in 25% of cases[40] when using 56 electrodes in a matrix configuration around the rat sciatic nerve. More recent advances are mainly associated with the FINE neural cuff electrode[41], which constrains the nerve into a rectangular cross-section. Modeling and tank studies suggested ~1 mm

localization accuracy using FINE electrodes with 16 contacts around the dog sciatic nerve[42–44]. Repeatability of fascicle imaging with ISA is also unclear. Wodlinger et al.[45] reconstructed source locations overlaid on histology in three of the seven nerves studied. Of the six fascicles represented, one was completely off, and three were largely outside the histological image of the fascicle.

In the brain, the highest SNR has been found empirically to be at 1.7 kHz. At lower frequencies, the impedance-evoked voltages are contaminated by endogenous potential activity, such as the EEG, evoked potentials, or seizure activity. At higher frequencies, the impedance change reduces as more applied current crosses through the neuronal membrane at rest, so there is less difference as ion channels open during activity. In this work, a higher applied frequency of 6 kHz was found to provide the highest SNR. This may be attributed to the higher component frequencies in the CAP and the differing biophysics of myelinated nerve fibers. The effect has been confirmed in a biophysical model of impedance changes during the CAP in peripheral nerve[26]. A concern when undertaking impedance measurement is that the applied current might itself cause changes in voltage sensitive ion channels and alter neuronal excitability. The current used in this work was 60 μA at 6 kHz. It was shown that current of up to double this value has a negligible effect on neuronal excitability; this was borne out in these studies, as there was no effect on the averaged CAP recorded[26].

The changes recorded in this study were only of larger myelinated A and B fibers. For use in autonomic nerve studies, this poses a limitation, as the majority are slow conducting C fibers. Similar impedance changes occur in unmyelinated nerve and have been demonstrated in the walking leg nerve of the crab[46]. The frequency for recording these is different and probably optimal at 225 Hz[47]. A challenge of EIT of these fibers is dispersion—the impedance change may become vanishingly small at >10 cm from the initiation site. This may pose a limitation to application in long unmyelinated nerve activity. However, recent modeling in our group has indicated that this too may be possible with sophisticated signal processing and longer averaging times[48].

In this work, averaging took 14 min for each image dataset (for each fascicle). This does not pose a problem if it is possible to stimulate peripheral branches continually for several minutes. However, in this work a SNR of ~17 [−] was produced. It may still be possible to produce good EIT images with shorter averaging. Depending on the application, it may well be practicable to collect data and average over longer periods of hours, if, as planned, a nerve cuff could be left in situ for recordings over hours or days. In addition, the SNR could be improved through spatiotemporal methods of activity extraction; for example,

velocity-selective[49] or variance-based methods, similar to those used for advanced ISA algorithms[44].

In this study, we have demonstrated the feasibility of using fast neural EIT for imaging fascicular activity within peripheral nerve with a high spatio-temporal resolution and cross-validated the resulting EIT images against neural tracer histology and microCT. This is encouraging to the view that this method might be used to identify fascicles of interest in human Electroceutical activity in the future. In the complex peripheral nerves of large diameter, the organ/function-specific fascicles might be organized into groups of fascicles, not a single fascicle per organ/function. However, it is expected that these fascicles are located in close proximity to one another. Branching of the putative organ-specific fascicles from the vagus nerve may occur in surgically inaccessible locations (e.g., within the thoracic cavity for cardiopulmonary projections) and thus branch-specific stimulation may not be feasible. Time difference fast neural EIT may then be achieved by creating datasets from different physiological states over time. This may be by triggering from spontaneous rhythmic activity or by extraneous induction of different states of physiological activity. Examples include activation of pulmonary stretch receptors (respiration-gated fascicular activity), cardiac baroreceptors (ECG-gated fascicular activity), esophageal distension, gastric distension, and nutrient-mediated activation of hepatic afferents. This could then enable selective stimulation of identified fascicles using the same nerve cuff. Imaging of slow conducting C fibers would require averaging over longer periods of time and alternative stimulation approaches such as box-car paradigm, where impedance changes are imaged between two windows: baseline and optimal continuous stimulation[50]. In this paradigm, the SNR can be increased by orders of magnitude because it allows application of low-pass temporal filtering over the recorded impedance signals before reconstruction. As the vision for the future is that this would occur with implanted devices, the requirement to average over hours should not be a limitation. Compared to the other in vivo techniques for imaging peripheral nerves, such as ultrasound, optical coherence tomography, or magnetic resonance tomography, a significant advantage of fast neural EIT is that it allows reconstruction of functional activity. The method could be used to monitor and adjust selective stimulation interactively over lengthy periods.

## Methods

All experiments were performed in accordance with the European Commission Directive 2010/63/EU (European Convention for the Protection of Vertebrate Animals used for Experimental and Other Scientific Purposes) and the UK Home Office Scientific Procedures Act (1986) with project approval from the University College London Institutional Animal Welfare and Ethical Review Committee.

**Neural tracing**. Adult male Sprague-Dawley rats (400–450 g) were anesthetized with isoflurane (5% induction, 2% maintenance). The tibial and peroneal branches of the sciatic nerve on both sides were exposed and pressure injected with 3 μL of 2.5% Alexa Fluor 488 or Alexa Fluor 555 dextran 10 kDa (ThermoFisher) in 0.01 M phosphate-buffered saline (PBS) using a glass micropipette ~2 cm distal to the branching region of the nerve. The speed of infusion was maintained at 0.5–1 μL/min and the tip of the pipette was left in a position for 2 min after completion of the injection to prevent tracer leakage. The animal recovered for 48 h before EIT recordings to ensure diffusion of the tracer to the level of the common sciatic nerve.

**In vivo preparation**. Animals were anesthetized with urethane (1.3 g/kg, i.p.), intubated, and artificially ventilated using a Harvard Apparatus Inspira Ventilator (Harvard Apparatus, Ltd, UK) with a 50/50% gas mixture of $O_2$ and air. Electrocardiogram and respiratory parameters (respiratory rate, end tidal $CO_2$) were monitored (Cardiocap 5, Datex Ohmeda). The core body temperature of the animal was controlled with a homeothermic heating unit (Harvard Apparatus, Kent, UK) and maintained at 37 °C. The animal was positioned prone and access to the common sciatic nerve was gained by dividing the biceps femoris and vastus lateralis muscles. Access to the posterior tibial nerve was established through dividing tibialis anterior and extensor digitorum longus muscles. The common peroneal nerve was accessed through a 2-mm lateral incision in biceps femoris near the knee

joint. Sural nerves were dissected from their origin from sciatic nerves to the lateral malleolus. Stimulation cuff electrodes (CorTec Gmbh, Freiburg, Germany) coated with poly(3,4-ethylenedioxythiophene):p-toluene sulfonate (PEDOT:pTS) were placed around the three branches, and an EIT cuff (see subsection "Fast neural EIT: electrode design") was placed around the main trunk of the sciatic nerve. After that, the retainers were removed, and surgical incision areas were closed with cotton pads soaked in 0.9% sterile saline solution. When the surgical procedure was completed, the animal was given pancuronium bromide (0.5 mg/kg, intramuscular) to avoid movement artifacts.

**Fast neural EIT: electrode design**. Flexible custom-made EIT cuff electrode arrays were made from laser cut stainless steel foil (12.5 μm thick) on medical-grade silicone rubber[29]. Arrays comprised 2 rings of 14 0.14 × 1 mm pads and 2 reference ring electrodes placed at extremities of the cuff. The 14 electrodes over each ring-like arrangement were equally spaced, thus radial distance between adjacent pads was ≈25.7°. The distance between electrode arrays was 2 mm (gap between centers), and the distance between each electrode array and closest reference electrode was 2.2 mm (edge to edge). Only one reference electrode and one 14-electrode ring was used for the recording EIT in these experiments (the second ring and second reference electrode are part of the design to allow compatibility of the cuff with other applications such as selective stimulation and imaging, which are not part of this publication). The cuff is designed to wrap around a nerve 1.4 mm in diameter. Electrodes were coated with PEDOT:pTS to reduce contact impedance and noise from the electrode–electrolyte interface[29]. To stabilize the cylindrical shape of the array and to secure the cuff around the nerve, the array was glued onto the inner side of a 1.5-mm internal diameter (I.D.) silicone rubber tube to form a 1.4-mm I.D. cuff, with thread sutured to assist cuff opening during implantation. The connector side was attached to a custom-printed circuit board adapter using 30 pin 1.5 mm connectors (Farnell, UK).

**Fast neural EIT: electrical stimulation of the fascicles and impedance measurements**. Recordings of the evoked activity in tibial, peroneal, and sural fascicles of the sciatic nerve were performed using the EIT cuff electrode array placed over the common part of the nerve. Branches were individually stimulated supramaximally at 1–2 mA amplitude, 5 Hz frequency, and 50 μs pulse width with CorTec cuff tunnel bi-polar electrodes (CorTec GmbH, Germany) placed ~1–1.5 cm distally from the EIT cuff (Fig. 1). A current of 1–2 mA was chosen to maximize the magnitude ratio between the CAP and the stimulus artifact measured on the EIT cuff electrode array prior to EIT recordings. The EIT protocol comprised 14 current injections (4-off spacing drive pattern, or ≈100°, Fig. 1b), each 60 s long, 300 trials averaged over each injection, with 14 min averaging in total (42 min for all 3 branches). The injected current frequency and amplitude was 6 kHz and 60 μA, respectively. Averaging over repeated stimulation pulses was required to reduce noise and reach an SNR sufficiently high for successful imaging. EIT measurements were performed with a ScouseTom system[31]. Raw signals were converted to δV recordings ("traces," Fig. 1b) by Hilbert transform demodulation (magnitude-only) at ±2 kHz bandwidth around the 6 kHz EIT carrier.

Criteria for exclusion of collected impedance traces from subsequent reconstruction process were as follows:

Injection/measurement on faulty electrode (characterized by extremely high noise compared to the rest of the electrodes).
DC saturation of raw signal.
δV background noise >3 μV.

**Fast neural EIT: image reconstruction**. EIT images were reconstructed using the following procedure[25,28,51].

The forward problem solution was computed using a complete electrode model within the UCL PEITS fast parallel forward solver[52]. The electrode contact impedance value in the solver was set to 1 KΩ and EIT current 60 μA. An extended forward model was used[28]; compared to the basic cylindrical model employed in our previously published proof-of-concept study[25], it included an additional external subdomain largely more conductive than the nerve itself. The new subdomain was included after observing that the original model simulated boundary voltages did not properly correlate with experimental ones due to the leakage of the electrical current at the edges of the cuff. A cylindrical model of the nerve with uniform baseline conductivity was set to 0.3 S/m, the external environment of the nerve was accounted for with higher conductivity (1.5 S/m), and cuff electrodes were placed over the external surface of the nerve. The conductivity of the external environment was chosen to be close to the conductivity of physiological solutions like 0.9% NaCl (1.4 S/m) and respective modeling studies[28]. This geometrical design was converted to a 2.63M-elements tetrahedral mesh for the purpose of computing the forward solution. The mesh element size was set to 20 μm on the electrodes, 40 μm for the inner nerve region, 60 μm for the outer part of the nerve region, and 420 μm for the external subdomain. The mesh quality was >0.7 for >99% of the elements[28]. The forward solution provided a linearized Jacobian matrix $J$ relating voltage traces δV measured at the electrodes to variations in conductivity (δσ): $δV = J \cdot δσ$[28].

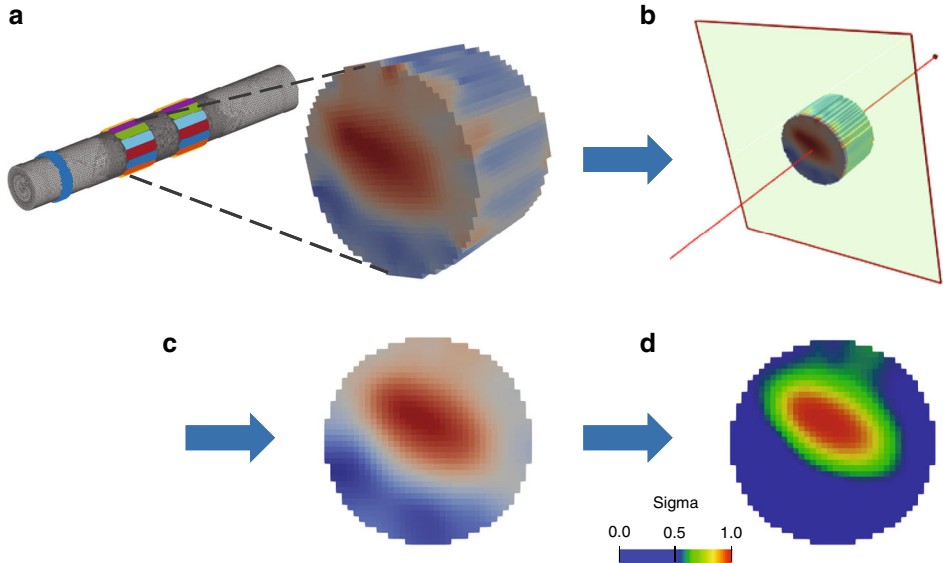

**Fig. 6 Post-processing of the reconstructed EIT image. a** Nerve subdomain of the 2.63M-elements tetrahedral mesh and zoom in of conductivity variation reconstructed over the coarse version of the electrode ring used for the EIT protocol. **b** Selection of the slice on the edge of EIT ring, on the side closest to the reference ring. **c** Slice-based median and mean filtering. **d** The range of values is normalized between 0 and 1, and the rainbow color scale indicates the top 50% of color for each image, i.e., full width at half maximum intensity scaling.

*Inverse problem*. To image local conductivity changes, a Jacobian matrix $J$ obtained from the forward solution was converted into a coarse hexahedral mesh of voxel size 40 μm and inverted using 0th-order Tikhonov regularization with noise-based voxel correction[25,53]. Only measurements on the single ring of electrodes of the EIT cuff (same ring where EIT current was injected) were used (Figs. 1b and 6a), for a total of 196 measurements (14 current injections × 14 electrodes), minus traces excluded according to the criteria listed in the section "Fast neural EIT: electrical stimulation of the fascicles and impedance measurements" above.

EIT reconstructions were evaluated at the time of peak average variations in measured voltage δV (Fig. 1b)[54,55]. Background inter-stimulus noise was also projected into the volume using the same reconstruction parameters. For each reconstruction, each voxel was divided by the standard deviation of projected noise, resulting in the *z*-score of the conductivity perturbation with respect to the background noise. Reconstruction was performed only on a relevant subset of ≈40K-elements on the hexahedral mesh, namely, the part of the nerve subdomain including the injection/measurement ring, as shown in Fig. 6b. Visualization of the reconstructed images was performed using Paraview (Version 5.7.0, Sandia National Labs, Kitware Inc., and Los Alamos National Labs, USA).

Images were evaluated at the cross-sectional slice on the edge of the EIT electrode array, as found to be optimal from simulations and previous studies[28]. Reconstructed slices were post-processed with median and mean filtering of 1- and 3-voxels radius, respectively (Fig. 2c). CoM was computed for each fascicle at full width at half maximum; for each slice, only the top 50% (magnitude) of voxels were included in the computation of metrics (Fig. 2d). This approach was chosen in order to avoid the evaluation being affected by the noise and small artifacts inherent to all the reconstructed images.

**Tissue collection and processing**. At the end of the in vivo experiment, the rat was euthanized and the sciatic nerves together with the branches (tibial, peroneal, and sural) were excised, fixed in 10% formalin (Sigma Aldrich), and stained with iodine for microCT scanning (see below). To enable co-registration, prior to removal of the EIT cuff, the opening area of the cuff was labeled; after removal of the cuff, a 6.0 surgical silk suture was glued to the epineurium to demarcate the cuff opening. The suture remained in place after all fixation and staining procedures (Fig. 4c). When microCT scanning was complete, the nerves were soaked in PBS to ensure washout of the excess of iodine solution. The nerves were then embedded in paraffin and transversely sectioned at 4 μm. Hematoxylin and eosin staining was performed on some sections corresponding to the area of EIT recordings, whereas some sections were cover-slipped with fluorescence-preserving medium for direct histological analysis of fluorescent neural tracers. Fluorescent images were obtained using a Leica microscope at ×10. The sural fascicle was located by visual inspection and identified by the absence of the fluorescent tracer, which labeled the peroneal and posterior tibial fascicles. Image analysis was performed in ImageJ (Version 1.52p, National Institutes of Health, USA).

**MicroCT scanning and image analysis**. Following fixation, the sciatic nerve was stained in 1% Lugol's iodine solution (total iodine 1%; 0.74% KI, 0.37% I, Sigma Aldrich) for 24 h. After staining, the nerve was blotted dry to remove excess iodine

solution and wrapped into a piece of cling film to avoid shrinkage of the nerve tissue during scanning. The nerve was then pulled in a taut position and fixed with a cylindrical sponge matrix within a custom-made three-dimensional (3D)-printed plastic mount, which was placed into a microCT scanner[12]. The microCT scanner (Nikon XT H 225, Nikon Metrology, Tring, UK) was homed and conditioned at 200 kVp for 10 min before scanning. The scanning parameters were optimized previously for highest image quality and contrast[12]. All nerves were scanned using a molybdenum target, a power of 4 W, 2903 projections, and a resolution with isotropic voxel size of 4 μm; the scanning parameters were set to 35 kVp energy, 114 μA current, and 4 s exposure time (0.25 frames per second). Scans were reconstructed using the Nikon CT Pro 3D software (Version XT 4.4.4, Nikon Metrology, Tring, UK). Center of rotation was calculated manually and beam hardening correction was performed with a preset of 2 and coefficient of 0.0. The reconstructions were saved as 16-bit volumes and triple TIFF 16-bit image stack files. Computerized 3D reconstruction and fascicle tracking was then performed on the reconstructed scans. The volume data were post-processed in MATLAB (Version r2018b, The MathWorks, Natick, USA)[12]. Subsequently, NIFTI files were produced to be compatible with Seg3D (Version Seg3D2-2.4.2, NIH Centre for Integrative Biomedical Computing, SCI Institute, University of Utah, USA) for visual inspection and segmentation of reconstructed microCT scans. Reconstructed microCT scan images were analyzed in ImageJ (Version 1.52p, National Institutes of Health, USA) in the *XY* plane to view the cross-section of the nerve[12].

**Image co-registration**. A total of eight rat sciatic nerves, from four animals, were used for the recordings of which five nerves were successful, yielding a full dataset with a combination of all three techniques. The other three nerves were excluded from the co-registration analysis due to failure of the neural tracing procedure in one of the fascicles or absence of the CAP or impedance signal. Of the five nerves used, four were on the right and one on the left side. The images from the single left sided nerve were flipped in the horizontal plane to allow for combination of all nerves in the same dataset. CoMs from EIT images were compared with CoMs from microCT scans and from histology slices with fluorescent neural tracing (referred to as "histology" here). Fascicular CoMs from reference microCT scans and histology images were evaluated by fitting the nerve external boundary to a circular profile after rigid image deformation and manual co-registration of the nerve cuff's opening (marked with the suture glued to the surface of the nerve) by image rotation (example of co-registration process is given in Fig. 4). CoMs from EIT, histology, and microCT were compared over vector distance modulus and over radial coordinates ($R/\vartheta$) calculated in MATLAB (Version r2018b, The MathWorks, Natick, USA).

**Statistics and reproducibility**. Impedance changes in four out of five nerves included in the study were taken in two repeats for each individual fascicle; all resulted in successful EIT image reconstructions. In the first nerve, repeats were not taken due to limited experimental time. Only one EIT image per fascicle was included in the resulting dataset (chosen based on the best SNR). Data analysis is performed with MATLAB. A paired *t* test was performed on peak δV traces with respect to background inter-stimulus noise to determine the significance of

impedance changes. Two-way ANOVA was used to evaluate the angular position of the CoM as a function of two independent factors: type of fascicle (tibial, peroneal, and sural) and measurement technique (EIT/tracers/microCT). Scatter of fascicle data clusters was computed for each fascicle type and technique as the standard deviation around mean fascicle position, along both coordinate axes. Resulting values were averaged to obtain mean scattering for each technique. Values are presented as mean ± 1 SD unless otherwise stated.

**Reporting summary**. Further information on research design is available in the Nature Research Reporting Summary linked to this article.

## Data availability

The data that support the findings of this study are available within the article and supplementary files or available from the corresponding author upon request. The raw EIT data are deposited in https://doi.org/10.5522/04/13177073.v1[55]. Source data are provided with this paper.

## Code availability

All scripts and processing code are available at: https://doi.org/10.5281/zenodo.4153151[28,54].

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

## Acknowledgements

This work was supported by the UK Medical Research Council (MRC grant No: MR/R01213X/1) and NIH SPARC (1OT2OD026545-01). A.V.G. is a Wellcome Trust Senior Research Fellow.

## Author contributions

E.R., S.M., and N.T. designed and performed the experiments, analyzed and interpreted the data and wrote the manuscript. F.I. and P.R.S. provided methodological and resource support for the microCT imaging. J.P., A.V.G., K.A., and D.H. supervised the study, interpreted the data, and revised the article.

## Competing interests

The authors declare no competing interests.
