## [Peer Review File · Nature Communications]

REVIEWER COMMENTS

Reviewer #1 (Remarks to the Author):

I think this is an impressive work. Especially, considering how small the scenario is, how small the impedance signal is, and the challenges in validating the process. The work seems well cited and put in context. Given their several developmental publications developing their procedure, it seems appropriate to have an article summarizing their more matured/developed technique.

I have a number of small comments, and one concept question.

Concept question: In this study, you first needed to stimulate the individual fascicles of interest (peroneal, tibial, and sural). Given these stimulations you have been successfully able to locate them in the sciatic nerve before they branch off via the EIT nerve cuff. If on patients you wanted to stimulate only an individual fascicle, then wouldn't just you do that directly, especially when this method would need that procedure to be performed anyway? As an engineer (not a clinician) reading this paper this point was not clear to me. If this could be made clearer perhaps in the 'Future Applications' subsection that I think could improve the paper's impact. Unless this is an obvious point I've missed.

Small points:

- In the abstract please I believe you should state 'nerve cuff' instead of cuff. I finished the abstract and was unable to determine if the EIT solution was invasive or not.
- Line 13, 'cause' side effects?
- Line 64, PNS is peripheral nerve system? I would guess this should be defined.
- Lines 113-117: I think this sentence needs a reference, is it shown in 23 or 24 or just hypothesized or predicted via a nerve models?
- Line 139: I think you should note the total collection time (is it 60 seconds x 14 electrode pairs = 14 minutes for each fascicle or 42 minutes for all three).
- Line 184, What is DiI?
- The blue arrow in the EIT image in Fig 1B could be labeled, like injected current.
- In Fig. 4b middle, what do you mean by rigid deformation? The figure looks squished in one direction and not so much in the other. I would general call that deformation. What part is rigid?
- Line 270 and line 373, is that 17 'dB'?
- Line 290, The claim that 'thus can provide a unique localized image' does not seem like a clear conclusion based on the previous sentence. I'd suggest adding a qualifier such as 'likely' or 'is expected to'
- Fig. 6 is there a colormap needed?
- Line 515-516, the 1.5 S/m correspond to a specific tissue or fluid or is it just an effective value that helps to reduce the mismatch.
- Unless I missed this, I did not see a reference to the supplemental results to see the full results from the 5 nerves. Can you add this?

Reviewer #2 (Remarks to the Author):

Ravagli and colleagues utilized a fast neural EIT technique to reconstruct images of sciatic nerve branches, namely peroneal, tibial and sural branch. The anatomical localization of the 3 branches was co-registered with images from microCT or fluorescent tracers. They matched well, thus validated the accuracy of EIT mapping. Overall, this is a helpful study. It establishes a potential non-invasive way to track peripheral nerve branches. Although the study did not go into details, the technique could provide interesting insights on nerve physiological activities on the EIT mapping.

I only have a few minor concerns listed below.

1. This paper is somewhat spoiled by a strong desire to “sale” its achievement of high resolution images. Throughout the manuscript, authors called the peroneal, tibial and sural “branches” as “fascicles”. However, EIT images clearly only can discern branches, not fascicles. Please see the definition of sciatic nerve “branch” (https://link.springer.com/chapter/10.1007/978-3-319-09522-6_10); and “fascicle” (<https://journals.plos.org/plosone/article/figure?id=10.1371/journal.pone.0200392.g004>) in the links. I understand that “fascicle” sounds a higher resolution, but you are there yet.
2. Please explain how the resolution of 200um was determined?
3. Nerve ultrasound can achieve a comparable resolution but will not provide any information of physiological activities. The later one is the strength of the EIT technique. Authors should elaborate on this subject. For instance, EIT images show much higher signal intensity at the center of every sciatic nerve branch, why? Does this mean there are higher density of myelinated nerve fibers toward the center of each branch? If yes, is there any histological evidence for this?
4. The manuscript started with a mission statement to solve the problem in the VNS therapy due to VNS lacking specificity to stimulate specific branch or fascicle in the vagus nerve. Well, vagus nerve is an autonomic nerve. Yet, toward the end of this paper, authors told the reader that this technique is limited to study autonomic nerve due to very slow conduction velocities in these non-myelinated nerve fibers. This makes readers very confused – mission is not accomplished?!
5. Line 317, it reads “is has no field cancellation...”. There is a typo (“it”). It should be “it has no field...”.

Manuscript ID: NCOMMS-20-20213-A

Responses to the referees' comments

We are grateful for the constructive comments we received from the reviewers of **Nature Communications** and have taken full account of the raised criticisms. We now provide a full response to all the reviewers' comments and submit the revised version of our manuscript.

Below we state the reviewers' points and then provide our responses.

REVIEWER #1

I think this is an impressive work. Especially, considering how small the scenario is, how small the impedance signal is, and the challenges in validating the process. The work seems well cited and put in context. Given their several developmental publications developing their procedure, it seems appropriate to have an article summarizing their more matured/developed technique. I have a number of small comments, and one concept question.

Response: We would like to thank this referee for his/her time taken to review our manuscript and positive assessment of our work.

Concept question: In this study, you first needed to stimulate the individual fascicles of interest (peroneal, tibial, and sural). Given these stimulations you have been successfully able to locate them in the sciatic nerve before they branch off via the EIT nerve cuff. If on patients you wanted to stimulate only an individual fascicle, then wouldn't just you do that directly, especially when this method would need that procedure to be performed anyway? As an engineer (not a clinician) reading this paper this point was not clear to me. If this could be made clearer perhaps in the 'Future Applications' subsection that I think could improve the paper's impact. Unless this is an obvious point I've missed.

Response: Thank you for this question. In clinical settings, it is not possible to electrically stimulate the branches of the vagus nerve as the branches may be in surgically inaccessible areas, and exposure of the organ-specific branches may cause unnecessary damage. For identification of organ/function-specific fascicles within the vagus nerve and their selective stimulation, other approaches will be used, such as creating EIT images by triggering from rhythmic spontaneous activity such as the ECG or respiration, or by changing the state of an organ, such as by inflating a balloon in the stomach. We revised the last paragraph of the Discussion in the updated manuscript to address this topic.

Added text: Lines 397-409

Branching of the putative organ-specific fascicles from the vagus nerve may occur in surgically inaccessible locations (e.g., within the thoracic cavity for cardiopulmonary projections) and thus branch-specific stimulation may not be feasible. Time difference fast neural EIT may then be achieved by creating datasets from different physiological states over time. This may be by triggering from spontaneous rhythmic activity, or by extraneous induction of different states of physiological activity. Examples include activation of pulmonary stretch receptors (respiration-gated fascicular activity), cardiac baroreceptors (ECG-gated fascicular activity), esophageal distension, gastric distension, and nutrient-mediated activation of hepatic afferents.

Small points:

• In the abstract please I believe you should state 'nerve cuff' instead of cuff. I finished the abstract and was unable to determine if the EIT solution was invasive or not.

Response: Thank you for pointing this out. We have since corrected the abstract to state "nerve cuff".

Original text: Line 4 of the Abstract

Fast neural electrical impedance tomography (EIT) allows fascicular CAP imaging with a resolution of <200 μm , <1 ms using a non-penetrating flexible cuff electrode array.

Amended text: Line 4 of the Abstract

Fast neural electrical impedance tomography (EIT) allows fascicular CAP imaging with a resolution of <200 μm , <1 ms using a non-penetrating flexible **nerve** cuff electrode array.

- Line 13, 'cause' side effects?

Response: Thank you, it has now been corrected.

Original text: Line 13

This may result in the activation of unwanted organs and **so** side effects, which limits therapeutic opportunities.

Amended text: Line 13

This may result in the activation of unwanted organs and **cause** side effects, which limits therapeutic opportunities.

- Line 64, PNS is peripheral nerve system? I would guess this should be defined.

Response: Thank you, the definition/abbreviation has been added in the preceding sentence.

Amended text: Lines 62-64

Neural tracing followed by histological examination has been successfully used to study neural connections within the central and peripheral nervous system (**PNS**).

- Lines 113-117: I think this sentence needs a reference, is it shown in 23 or 24 or just hypothesized or predicted via a nerve models?

Response: Thank you, the reference has been added.

Amended text: Lines 112-118

The principle is that impedance in neuronal membranes falls as ion channels open during evoked activity ^(Holder, 2005). This produces a decrease in the bulk electrical impedance of ~0.1% during neuronal depolarization, which allows the applied EIT current to pass into the intracellular space, whereas at rest the EIT current predominantly travels in the extracellular space ^(Holder, 2005).

- Line 139: I think you should note the total collection time (is it 60 seconds x 14 electrode pairs = 14 minutes for each fascicle or 42 minutes for all three).

Response: Indeed, the total time of the recording is 14 min per fascicle (42 min for all three fascicles/branches of the sciatic nerve). This has been added to the general description of the fast neural EIT principle (line 133), discussion (physiological considerations, line 374) and methods section (line 516).

Amended text: Lines 130-133

A full dataset for image reconstruction is then collected by sequential switching between all possible 14 electrode pairs with the same spacing, using electronic multiplexers (**total time 14 min per branch/fascicle**).

Amended text: Lines 373-374

In this work, averaging took 14 minutes for each image data set (**for each fascicle**).

Amended text: Lines 513-517

The EIT protocol comprised 14 current injections (4-off spacing drive pattern, or $\approx 100^\circ$, Fig. 1b), each 60 s long, 300 trials averaged over each injection, with 14 min averaging in total (**42 min for all three branches**).

- Line 184, What is Dil?

Response: Dil stands for a passive lipophilic neural tracer 1,1'-dioctadecyl-3,3,3'-tetramethyl-indocarbocyanine perchlorate. The definition has now been given in the text, thank you.

Amended text: Lines 184-187

FG and **1,1'-dioctadecyl-3,3,3'-tetramethyl-indocarbocyanine perchlorate (Dil)** allowed reproducible labelling of the fascicles but impaired evoked CAPs, which obviated reproducible EIT data.

- The blue arrow in the EIT image in Fig 1B could be labeled, like injected current.

Response: Thank you for this comment, Figure 1 has since been corrected as per suggestion.

- In Fig. 4b middle, what do you mean by rigid deformation? The figure looks squished in one direction and not so much in the other. I would general call that deformation. What part is rigid?

Response: Thank you for this critique, the description was corrected for as follows: “single-axis rescaling”.

Original text: Lines 8-9 of Figure 4 legend

Middle: rigid deformation

Amended text: Lines 8-9 of Figure 4 legend

Middle: single-axis rescaling.

- Line 270 and line 373, is that 17 ‘dB’?

Response: ‘17’ in this instance is a measure of signal-to-noise ratio in arbitrary units (pure ratio, dimensionless), this corresponds to 24.6 dB. We updated the text with the symbol for dimensionless quantities [-] to make this clearer.

Amended text: Lines 270-272

Evoked impedance changes had a signal-to-noise ratio of 17 [-] for the imaged A and B fast myelinated fibers, applying 6 kHz current.

Amended text: Lines 376-377

However, in this work a SNR of ~17 [-] was produced.

- Line 290, The claim that ‘thus can provide a unique localized image’ does not seem like a clear conclusion based on the previous sentence. I’d suggest adding a qualifier such as ‘likely’ or ‘is expected to’.

Response: Thank you, the phrase has been corrected as per suggestion.

Original text: Lines 295-298

It **thus can** provide a unique localized image of neuronal depolarization but it is not necessarily restricted to individual fascicles.

Amended text: Lines 295-298

It **is likely to** provide a unique localized image of neuronal depolarization but it is not necessarily restricted to individual fascicles.

- Fig. 6 is there a colormap needed?

Response: Thank you, the colormap has been added, both to Figure 6d and Figure 4a.

- Line 515-516, the 1.5 S/m correspond to a specific tissue or fluid or is it just an effective value that helps to reduce the mismatch.

Response: The conductivity of the external environment of the nerve is set to 1.5 S/m to account for the average conductivity of extracellular space. For reference, physiological solutions such as 0.9% NaCl saline are similar in ionic concentration to extracellular space and have an average conductivity value of 1.4 S/m. The explanation has been added to the text.

Added text: Lines 549-552

The conductivity of the external environment was chosen to be close to the conductivity of physiological solutions such as 0.9% NaCl (1.4 S/m) and respective modeling studies ^(Ravagli *et al.*, 2019).

- Unless I missed this, I did not see a reference to the supplemental results to see the full results from the 5 nerves. Can you add this?

Response: Thank you for this comment, the reference to Supplementary Fig. 2 has been added in lines 252 and 258.

Amended text: Lines 248-251

Three clearly distinct zones of activation due to stimulation of tibial, peroneal and sural fascicles were apparent in reconstructed EIT images for each nerve (Fig. 4a, **Supplementary Fig. 2**).

Amended text: Lines 255-257

Visual inspection indicates a close correlation of the three fascicles across different techniques. (Fig. 4, Fig. 5, **Supplementary Fig. 2**).

REVIEWER #2

Ravagli and colleagues utilized a fast neural EIT technique to reconstruct images of sciatic nerve branches, namely peroneal, tibial and sural branch. The anatomical localization of the 3 branches was co-registered with images from microCT or fluorescent tracers. They matched well, thus validated the accuracy of EIT mapping. Overall, this is a helpful study. It establishes a potential non-invasive way to track peripheral nerve branches. Although the study did not go into details, the technique could provide interesting insights on nerve physiological activities on the EIT mapping.

Response: We would like to thank this referee for his/her time taken to review our manuscript and overall positive assessment of our work.

I only have a few minor concerns listed below.

1. This paper is somewhat spoiled by a strong desire to “sale” its achievement of high resolution images. Throughout the manuscript, authors called the peroneal, tibial and sural “branches” as “fascicles”. However, EIT images clearly only can discern branches, not fascicles. Please see the definition of sciatic nerve “branch” (https://link.springer.com/chapter/10.1007/978-3-319-09522-6_10); and “fascicle” (<https://journals.plos.org/plosone/article/figure?id=10.1371/journal.pone.0200392.g004>) in the links. I understand that “fascicle” sounds a higher resolution, but you are there yet.

Response: Indeed, in the human sciatic nerve, the organization of axons is different to the rat sciatic nerve, and in the paper you mentioned (Reina *et al.*, 2014), the fascicular organization of the human sciatic nerve is described, with tibial, peroneal, and sural nerves consisting of fascicular groups, not individual fascicles. The fact that this organization (number of fascicles) is different in various species is also mentioned in the second suggested paper (Troiani *et al.*, 2019) on the example of *Xenopus Laevis* (Figure 4, Troiani *et al.*, 2019: the sciatic nerve in this species is made of a single bundle of axons, or fascicle, surrounded by the perineurium and loose epineurium). In our study, we used the rat sciatic nerve because of its clear somatotopic organization (Badia *et al.*, 2010: <https://doi.org/10.1002/mus.21652>). In the rat sciatic nerve, both afferent and efferent nerve fibers at the level of common sciatic nerve (where the EIT nerve cuff electrode placed in our study) are organized in fascicles which, distally to the nerve cuff, give rise to common sciatic, peroneal and sural branches (Badia *et al.*, 2010). Therefore, we believe that our statement that fast neural EIT can be a means of imaging fascicular organization of the peripheral nerve is not an overstatement (in case of rat sciatic nerve, where one fascicle in the proximal part of the nerve corresponds to one branch in its distal part, please see microCT segmentation examples from Figure 3 in our paper, as well as tracing and reconstruction data on rat sciatic nerve from Badia *et al.*, 2010). In the “Future applications” section of the paper, we say “this method might be used to identify fascicles of interest in human Electroceutical activity in the future” (lines 390-392). We do appreciate the possibility that in the complex peripheral nerves of large diameter (e.g., vagus nerve) the organ/function-specific nerve fibers might be organized to groups of fascicles, not a single fascicle per organ/function. We added a respective paragraph to the Discussion section.

Added text: Lines 400-404

In the complex peripheral nerves of large diameter, the organ/function specific fascicles might be organized into groups of fascicles, not a single fascicle per organ/function. However, it is expected that these fascicles are located in close proximity to one another.

2. Please explain how the resolution of 200um was determined?

Response: Thank you for this question. Based on simulations confirmed by experiments in Ravagli *et al.*, 2019, centre-of-mass localization error was $172 \pm 46 \mu\text{m}$ for tibial fascicle, $110 \pm 52 \mu\text{m}$ for peroneal fascicle, and $141 \pm 56 \mu\text{m}$ overall. The corresponding reference is now added to the text (line 141).

Added text: Lines 138-141

The resulting dataset allows imaging of neuronal depolarization with a high spatio-temporal resolution of $<1 \text{ ms}$ and $200 \mu\text{m}$ in rat sciatic nerve (Aristovich *et al.*, 2018; Ravagli *et al.*, 2019).

3. Nerve ultrasound can achieve a comparable resolution but will not provide any information of physiological activities. The later one is the strength of the EIT technique. Authors should elaborate on this subject. For instance, EIT images show much higher signal intensity at the center of every sciatic nerve branch, why? Does this mean there are higher density of myelinated nerve fibers toward the center of each branch? If yes, is there any histological evidence for this?

Response: Thank you for this question and advice. In mathematical terms, EIT is a soft-field problem; therefore, the resulting images cannot have sharp borders due to the type of image reconstruction process needed. In this specific application, sharp edges from the real fascicles are blurred into the smooth transitions with a kernel size of $\sim 100\mu\text{m}$. This is not an issue for our intended use of nerve EIT as coordinates of the center point of the fascicle are the only information needed for targeted stimulation.

Added text: Lines 288-295

Here, we present the first validation of fast neural EIT in peripheral nerve. EIT is a soft-field imaging modality, with the resulting images having no sharp borders due to the type of image reconstruction process. In this specific application of fast neural EIT, the sharp edges from the real fascicles are blurred into the smooth transitions with a kernel size of $\sim 100\mu\text{m}$ and so typically fast neural EIT has a spatial resolution of $\sim 10\%$ of the image diameter.

Added text: Lines 421-425

Compared to the other *in vivo* techniques for imaging peripheral nerves, such as ultrasound, optical coherence tomography, or magnetic resonance tomography, a significant advantage of fast neural EIT is that it allows reconstruction of functional activity.

4. The manuscript started with a mission statement to solve the problem in the VNS therapy due to VNS lacking specificity to stimulate specific branch or fascicle in the vagus nerve. Well, vagus nerve is an autonomic nerve. Yet, toward the end of this paper, authors told the reader that this technique is limited to study autonomic nerve due to very slow conduction velocities in these non-myelinated nerve fibers. This makes readers very confused – mission is not accomplished?!

Response: Thank you for this comment. Indeed, this is just a proof-of-principle study, and further work needs to be done to enable this method to image slow conducting fibers. Our current work in progress is utilizing physiologically-gated activity instead of electrical stimulation. We already have some preliminary success in using the respiration-gated and ECG-gated activity with successful image reconstruction, where we extract the characteristic envelope of the impedance change in the frequency domain, and reconstructing the images of the averaged envelope. We also have had preliminary success in using the box-car paradigm, where we are imaging the impedance changes between two windows: baseline, and optimal continuous stimulation. In this paradigm, the SNR is increased by orders of magnitude because we can employ low-pass temporal filtering over the recorded impedance signals before reconstruction. We have added a paragraph to the Discussion where we have outlined the above methods as a potential means for imaging slow-conducting and spontaneous activity within the autonomic nerves.

Added text: Lines 397-419

Branching of the putative organ-specific fascicles from the vagus nerve may occur in surgically inaccessible locations (e.g., within the thoracic cavity for cardiopulmonary projections) and thus branch-specific stimulation may not be feasible. Time difference fast neural EIT may then be achieved by creating datasets from different physiological states over time. This may be by triggering from spontaneous rhythmic activity, or by extraneous induction of different states of physiological activity. Examples include activation of pulmonary stretch receptors (respiration-gated fascicular activity), cardiac baroreceptors (ECG-gated fascicular activity), esophageal distension, gastric distension, and nutrient-mediated activation of hepatic afferents. This could then enable selective stimulation of identified fascicles using the same nerve cuff. Imaging of slow conducting C-fibers would require averaging over longer periods of time and alternative stimulation approaches such as the box-car paradigm, where impedance changes are imaged between two windows: baseline and optimal continuous stimulation^(Macey 2016). In this paradigm, the SNR can be increased by orders of magnitude because it allows application of low-pass temporal filtering over the recorded impedance signals before reconstruction.

5. Line 317, it reads “is has no field cancellation...”. There is a typo (“it”). It should be “it has no field...”.

Response: Thank you for spotting this, it has since been corrected (line 321).

Reviewer #2 (Remarks to the Author):

Authors have addressed my concerns but not for my 1st comment. Fascicle versus branch are two clearly distinctive terms. This technique only discerns branches, not fascicles. The excessive justifications for this issue by the authors do not make sense at all. Please remove the statement of fascicle.

Responses to the referees' comments

We are grateful for the constructive comments and overall positive assessment of our work we received from the reviewers of *Nature Communications*. Below please find our response to the final comments of the Reviewer 2.

REVIEWER #2 (Remarks to the Author):

Authors have addressed my concerns but not for my 1st comment. Fascicle versus branch are two clearly distinctive terms. This technique only discerns branches, not fascicles. The excessive justifications for this issue by the authors do not make sense at all. Please remove the statement of fascicle.

Response

We appreciate the difference between rat and human sciatic nerve architecture, with the nerve fibers in the rat sciatic nerve organized into much smaller number of fascicles than in the human nerve. We believe that in the context of rat sciatic nerve anatomy we can talk about resolution on the fascicular level, and we changed the title of the manuscript to address this issue, as per Editor's suggestion. We have also added a paragraph discussing the differences between the species (lines 172-180).

Previous title:

Imaging fascicular organization of **peripheral** nerves with fast neural Electrical Impedance Tomography (EIT)

Amended title:

Imaging fascicular organization of **rat sciatic** nerves with fast neural Electrical Impedance Tomography (EIT)

Added Text: Lines 172-180

We chose the model of rat sciatic nerve for our study because of its clear somatotopic organization. Compared to the human sciatic nerve, where tibial, peroneal and sural branches consist of fascicular groups, not individual fascicles (Reina et al., 2014), the rat sciatic nerve is a simpler model of peripheral nerve. In rats, both afferent and efferent nerve fibers at the level of common sciatic nerve are organized into three main fascicles. These fascicles give rise to sciatic, peroneal and sural branches (Badia et al., 2010).